# Validating Salinity from SMAP and HYCOM Data with Saildrone Data during EUREC⁴A-OA/ATOMIC

Kashawn Hall [1,*], Alton Daley [1], Shanice Whitehall [1], Sanola Sandiford [1] and Chelle L. Gentemann [2]

1 Caribbean Institute for Meteorology and Hydrology, Husbands, St. James BB23006, Barbados; adaley@cimh.edu.bb (A.D.); swhitehall@cimh.edu.bb (S.W.); ssandiford@cimh.edu.bb (S.S.)
2 Farallon Institute, Petaluma, CA 94952, USA; cgentemann@faralloninstitute.org
* Correspondence: khall@cimh.edu.bb; Tel.: +1246-547-7283

**Abstract:** The 2020 'Elucidating the role of clouds-circulation coupling in climate-Ocean-Atmosphere' (EUREC⁴A-OA) and the 'Atlantic Tradewind Ocean-Atmosphere Mesoscale Interaction Campaign' (ATOMIC) campaigns focused on improving our understanding of the interaction between clouds, convection and circulation and their function in our changing climate. The campaign utilized many data collection technologies, some of which are relatively new. In this study, we used saildrone uncrewed surface vehicles, one of the newer cutting edge technologies available for marine data collection, to validate Level 2 and Level 3 Soil Moisture Active Passive (SMAP) satellite and Hybrid Coordinate Ocean Model (HYCOM) sea surface salinity (SSS) products in the Western Tropical Atlantic. The saildrones observed fine-scale salinity variability not present in the lower-spatial resolution satellite and model products. In regions that lacked significant small-scale salinity variability, the satellite and model salinities performed well. However, SMAP Remote Sensing Systems (RSS) 70 km generally outperformed its counterparts outside of areas with submesoscale SSS variation, whereas RSS 40 km performed better within freshening events such as a fresh tongue. HYCOM failed to detect the fresh tongue. These results will allow researchers to make informed decisions regarding the most ideal product and its drawbacks for their applications in this region and aid in the improvement of mesoscale and submesoscale SSS products, which can lead to the refinement of numerical weather prediction (NWP) and climate models.

**Keywords:** saildrone; salinity; Soil Moisture Active Passive (SMAP); Hybrid Coordinate Ocean Model (HYCOM); EUREC⁴A; ATOMIC; physical oceanography; remote sensing; air-sea interactions

## 1. Introduction

Sea surface salinity (SSS) influences ocean-atmosphere interactions which control weather and climate patterns. Understanding ocean-atmosphere interactions in the western tropical Atlantic is essential to the development of new parameterizations schemes for numerical weather prediction (NWP) and climate models [1] that support forecasting over this region. The need to improve weather and climate prediction models over this region is driven by the fact that many Caribbean Small Island Developing States (SIDS) located in the region heavily utilize NWPs to support severe weather forecasting as part of their multi-hazard early warning system and climate models to drive their climate change adaptation policies and programmes.

Studies have shown that salinity stratification as a result of riverine outflows from large South American river systems, in particular the Amazon River system, that make their way into the Northern Atlantic Ocean, reduce upper ocean cooling during hurricane passage supporting hurricane intensification [2,3]. SSS modulates both vertical mixing and Sea Surface Temperature (SST) [1]. The Barrier Layer (BL) produced as a result of freshwater influxes may produce biases in SST by controlling vertical mixing and entrainment of cooler water into the Ocean Mixed Layer (OML) [4]. The freshwater fluxes described also carry

nutrients and organic matter which can produce ecological challenges that impact fisheries, marine biodiversity and tourism [5].

Observations of ocean salinity and density provide information on ocean dynamics especially in areas with complex ocean processes such as the North Brazil Current (NBC) region [6–8]. Salinity variations provide insight into ocean circulation and air sea fluxes which in turn impact the atmospheric boundary layer, resulting in changes to the Earth's climate [9]. Salinity is also used for monitoring the movements of water masses along with vertical exchange of water between surface and subsurface layers [10]. Historically, in situ measurements of salinity and temperature were taken from ships using bucket and thermosalinograph (TSG) measurements and were sparse on a global scale [11]. This improved over time with the implementation of the Argo Float programme, which included in situ observations of salinity [12]. This global network, consisting of autonomous Argo Profiling floats produces 3° spatial resolution salinity data at 10-day intervals. Although useful for many applications, these spatial and temporal resolutions are too coarse to provide useful salinity observations at the mesoscale level [13].

In 2010, the first global satellite ocean salinity measurements became available from the European Space Agency's (ESA) Soil Moisture and Ocean Salinity (SMOS) mission, which retrieved salinity at a spatial resolution of 40 km and a 23-day repeat cycle. In 2011, National Aeronautics and Space Administration's (NASA) Aquarius mission began providing salinity at a spatial resolution of 100 km and a 7-day repeat cycle [14]. This was followed by NASA's Soil Moisture Active Passive (SMAP) mission in 2015, which has a resolution of 40 km and a repeat cycle of 8 days [15]. Although space-based platforms improved the acquisition of salinity data, there are still inaccuracies when making measurements near to the coast and at high latitudes, due to limitations of the L-band microwave radiometers used to measure salinity [16].

Ocean modelling provides another approach for estimating global ocean salinity. Ocean modelling commenced in the 1960s with the development of simple ocean circulation models [17]. These older models were incapable of simulating ocean processes at spatial resolutions finer than 100 km and many typically had temporal resolutions of approximately three months. However, improvements over the years have led to the development of models capable of simulating submesoscale ocean behavior. The accuracy of an ocean simulation model depends on how well the model represents ocean dynamics, the quantity and quality of input data as well as its ability to simulate air sea interactions among other things [18].

The HYbrid Coordinate Ocean Model (HYCOM) (https://hycom.org, accessed on 17 September 2021), used in this study, assimilates Argo float and Conductivity, Temperature and Depth (CTD) data for its salinity estimates. HYCOM has exhibited large errors near river discharge areas in the tropics and in boundary currents such as the Brazil-Mailvinas confluence and the Gulf Stream [19]. These errors may make the model a less than ideal choice for estimating SSS values in tropical discharge regions. Saildrone data can provide vital information regarding the extent of these errors in these locations and aid in the improvement of model output. Contemporary, high resolution in situ observations by saildrone uncrewed vehicles can resolve mesoscale and submesoscale variability observed in coastal regions and remain accurate at high latitudes, providing valuable data for process studies and validation of the satellite measurements [20].

The 'Elucidating the role of clouds- circulation coupling in climate-Ocean-Atmosphere component' (EUREC$^4$A-OA) and the 'Atlantic Tradewind Ocean-Atmosphere Mesoscale Interaction Campaign' (ATOMIC, US) campaigns resulted in an unmatched observing effort in the western tropical Atlantic. In this campaign, three autonomous saildrone vehicles collected data over a 45-day period in regions east of Barbados (Figure 1). SSS collected by the saildrones during the campaign are used to assess the validity of SSS data retrieved from satellites and HYCOM products. Satellite SSS products as well as HYCOM model outputs are used extensively in areas that are lacking in situ measurements such as the

region where the North Brazil Current occurs. Furthermore, HYCOM outputs are used as reference salinity fields for the calibration in SMAP products [21,22].

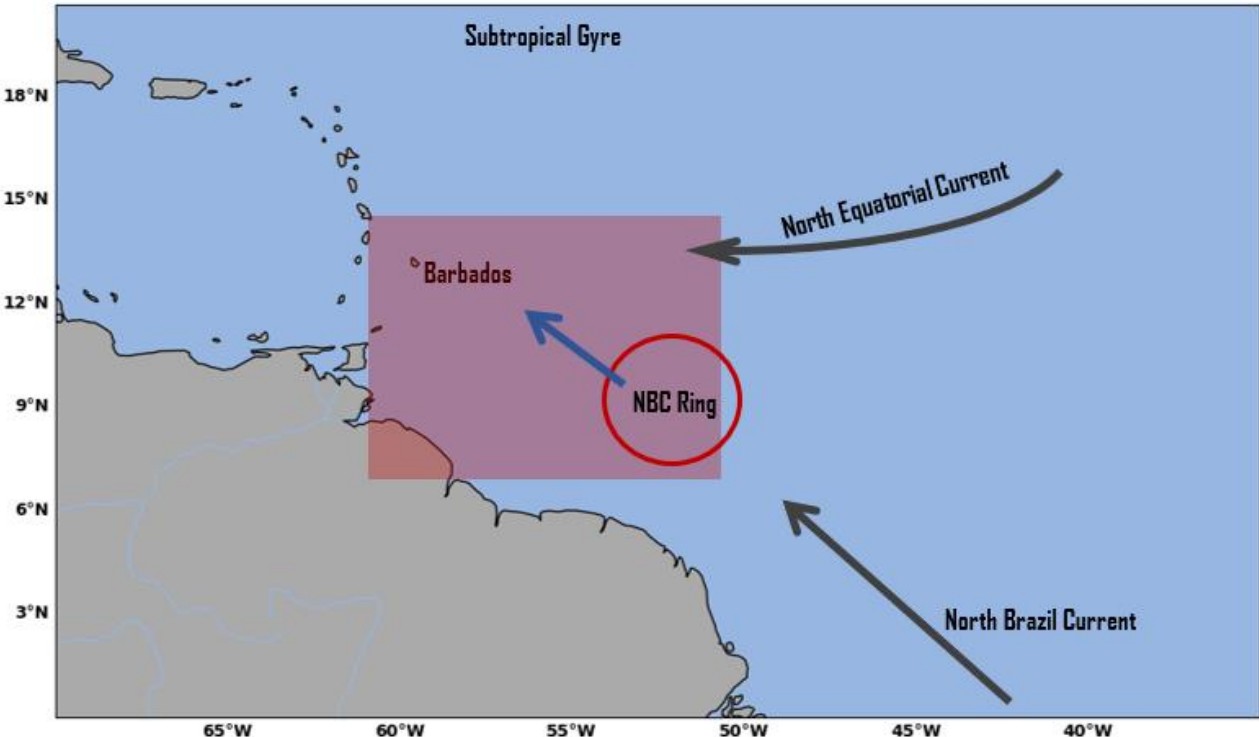

**Figure 1.** The region of EUREC⁴A-OA/ATOMIC study. Saildrones focused on the region with submesoscale variability, North Brazil Current (NBC) region (located in the red box), where the NBC rings (red circle) can result in strong fronts and ocean barrier layers. The blue arrow shows the direction of water transport via the NBC towards the Subtropical Gyre.

This study validates SMAP SSS products at both 70 km and 40 km resolution as well as the HYCOM estimated SSS in the Western Tropical Atlantic. The validation is carried out using saildrone SSS data collected during the 45-day EUREC⁴A-OA/ATOMIC campaign. The paper is presented in the following manner. Section 2 introduces the EUREC⁴A campaign and the marine data collection platforms, the SMAP products, HYCOM and their respective datasets collocation and validation methodologies. Section 3 presents the results of the comparison between the satellite products, HYCOM, and saildrones datasets. Additionally, Section 3 highlights the investigation of a fresh tongue encountered during the campaign. Finally, the results are discussed in Section 4.

## 2. Data and Methods

### 2.1. EUREC⁴A Campaign

The EUREC⁴A campaign was a cloud- and climate-focused undertaking, seeking to measure microphysical properties of trade-wind cumuli as a function of the large-scale environment and provide benchmark data for future satellite and modelling efforts [23,24]. Initially, a Caribbean-French-German partnership, the EUREC⁴A campaign, operated eastward of the island of Barbados in the lower Atlantic trades (Figure 2) and incorporated the Barbados Cloud Observatory (BCO), along with aircraft, research vessels, buoys and drifters. Surface level observations carried out by the research vessel Meteor and the BCO would serve to complement the airborne measurements within the 'EUREC⁴A-Circle' (Study area A east of Barbados where the HALO (High Altitude and Long Range Research Aircraft) would circle in range of the land-based radar PoldiRad (Polarization Diversity Doppler Radar), Figure 2) characterizing the atmospheric environment from the surface to provide full measurement coverage of the atmospheric column.

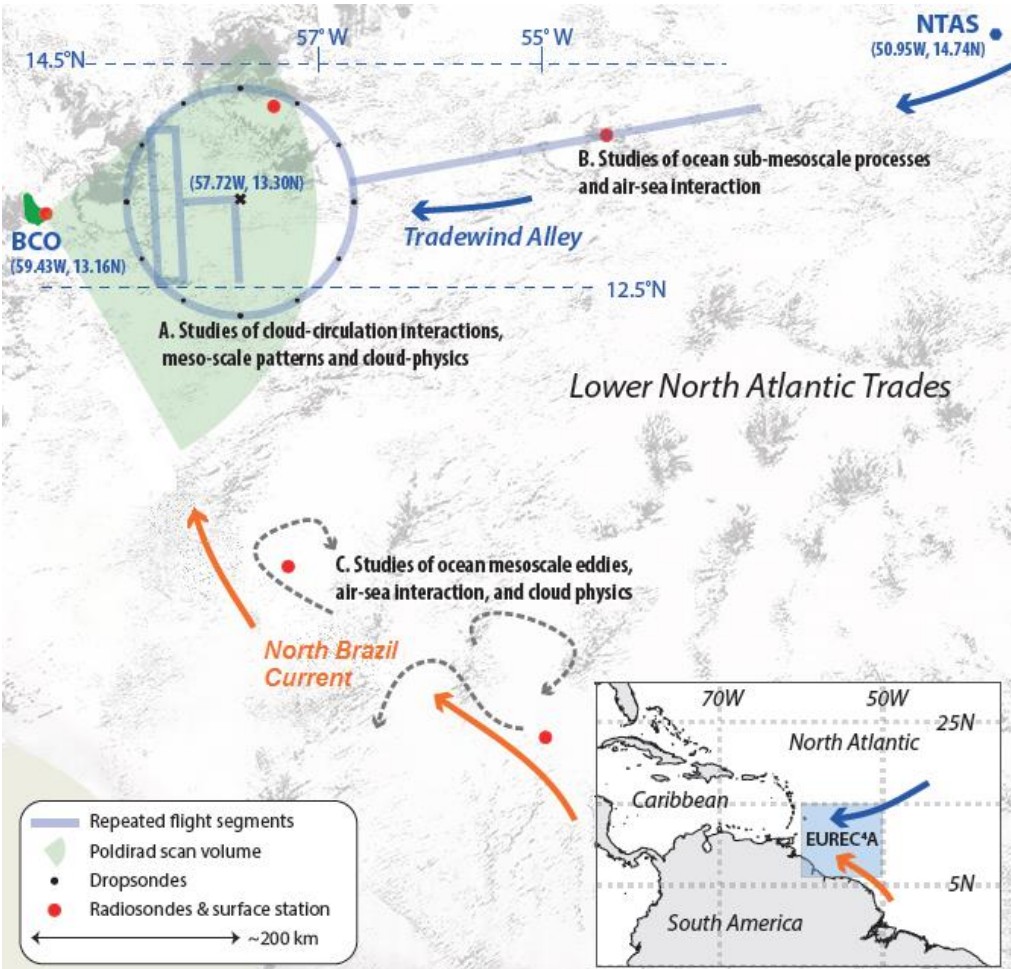

**Figure 2.** The EUREC[4]A study areas (**A**) located within the "EUREC[4]A-Circle", (**B**) within the 'Tradewind Alley' between the Northwest Tropical Atlantic Station (NTAS) and the Barbados Cloud Observatory (BCO) and (**C**) the North Brazil Current region. The inset map shows the location of the EUREC[4]A study area within the Atlantic Ocean. The background cloud field is taken from the 5 February 2020 MODIS-Terra overpass. Reprinted from Stevens et al., 2021 [23].

EUREC[4]A's aim eventually expanded to include the examination of how air–sea interaction is influenced by mesoscale eddies, sub-mesoscale fronts and filaments in the vicinity of the NBC [23]. Collaboration between the Ocean Atmosphere component of EUREC[4]A (EUREC[4]A-OA) and the ATOMIC campaign led to the inclusion of three additional research vessels (R/Vs): L'Atalante, Maria S. Merian and Ronald H. Brown. These vessels were fitted with instrumentation to measure both the atmosphere and the upper kilometer of the subsurface water column. Measurements from autonomous vehicles including ocean gliders, drifters, floats and saildrones were used to supplement measurements directly from the research vessels. More details of the motivations, the wide range of measurement platforms and instrumentations involved in the EUREC[4]A campaign can be read at Stevens et al. [23].

### 2.2. Saildrone

Saildrones are zero-emission, solar powered, wind propelled marine vehicles produced by Saildrone Inc. with science-grade instrumentation. They have 12 sensors to measure data at the air–sea interface (e.g., SSS, SST, air temperature, 3D winds) and can carry one additional sensor on the keel (either an Acoustic Doppler Current Profiler (ADCP) or scientific echo-sounders). A Global Positioning System (GPS) and an onboard computer enables the vehicles to navigate following prescribed waypoints, while staying within

a set corridor, taking winds and currents into consideration autonomously. Waypoints can be dynamically updated as environmental conditions change or interesting features develop. Vehicles are controlled and data transferred in near-real-time via two-way Iridium satellite communications. Saildrones can travel approximately 100 km per day, depending on wind speed, and can operate in remote regions where research vessels may be time- and cost-prohibitive. During the EUREC[4]A-OA/ATOMIC cruise, each saildrone vehicle measured salinity via the SeaBird-37-SMP-ODO Microcat and RBR CTD/ODO/Chl-A instruments at 0.5 m depth for 12 s, each minute.

This study analyzed saildrone data from the three NASA-funded saildrones (SD1026, SD1060 and SD1061) deployed during the EUREC[4]A-OA/ATOMIC campaign, which operated during the period 17 January–2 March 2020 within the domain 7–13.5°N and 48–60°W (Figure 3).

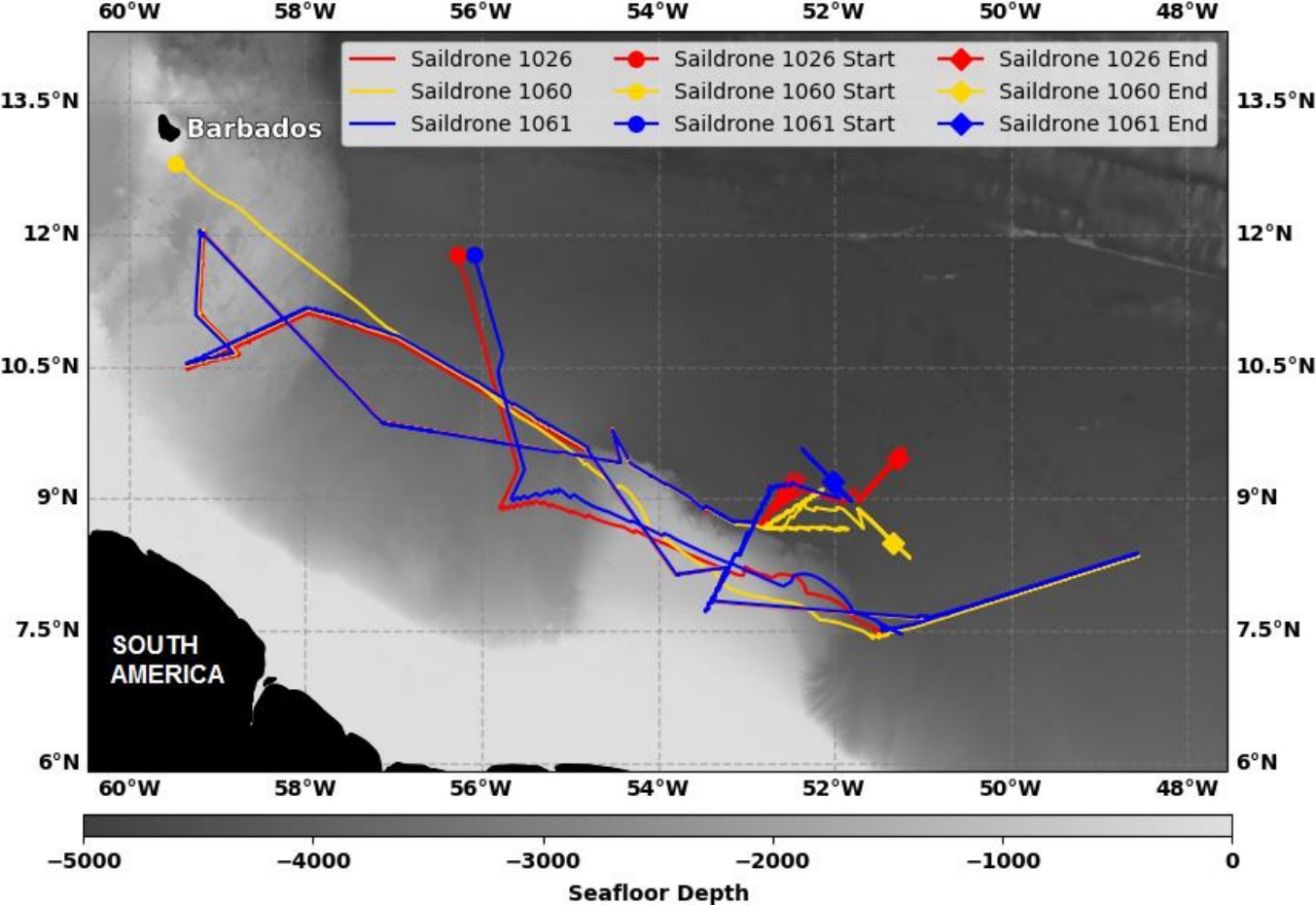

**Figure 3.** The tracks of the three NASA saildrones within the EUREC[4]A-OA study area. The start (17 January 2020) and end (2 March 2020) points of each track are shown by a circle and diamond, respectively. Bathymetry Data from The General Bathymetric Chart of the Oceans (GEBCO) (available at https://download.gebco.net/ (accessed on 27 June 2022) is shown behind the saildrone tracks.

The saildrone data files collected at a 1 min temporal resolution over the duration of the measurement period include platform telemetry and near-surface observational data (see https://podaac.jpl.nasa.gov/dataset/SAILDRONE_ATOMIC, accessed on 30 April 2021). The utilization of the saildrone platform represents a new and evolving trend in scientific sampling campaigns in which the data collection platform and process are subcontracted to Saildrone Inc. who is responsible for executing the experiment design and assuring the quality of the data. Hence, it is important to independently assess the accuracy of the data provided by the Saildrone operators.

Preceding the EUREC[4]A-OA/ATOMIC campaign, saildrone-measured SSS data were validated in polar latitudes during the Innovative Technology for Arctic Exploration (ITAE) by Cokelet et al. [25] and in the tropics during the Salinity Process in the Upper-ocean Regional Study 2 (SPURS-2) campaign in 2017 [26]. SSS for the saildrones during the polar and tropical campaign had a root mean squared difference (RMSD) of 0.01 and 0.0075 practical salinity unit (psu), respectively.

Similar to our current study, SSS and SST data from saildrones were used to validate SMAP satellite derived SSS and SST datasets during a 2018 Baja California campaign. Much like the NBC region in the present study, the California current system is dominated by submesoscale and mesoscale variability. Vazquez-Cuervo et al. [20] found that saildrone SSS showed fresher biases compared to the SMAP products.

### 2.3. The Soil Moisture Active Passive (SMAP) Data

The SMAP mission is a polar orbiting, remote sensing observatory developed by the NASA primarily for soil moisture mapping. SMAP uses an L-band radar and radiometer with central frequencies of 1.26 GHz and 1.41 GHz, respectively. SMAP primarily detects brightness temperature, which is then converted to soil moisture on land and SSS in the ocean due to the L-band microwave's sensitivity to salinity. SMAP data is independently processed by the NASA Jet Propulsion Laboratory (JPL) and the Remote Sensing System (RSS). Both data providers produce Level-2 (L2) data, which contain data from a single orbit of the satellite (orbital data), and Level-3 (L3) data which are 8-day averages of SMAP salinity data. This study uses SMAP JPL version 5, L2 and L3 data at a spatial resolution of 60 km from 17 January to 2 March 2020 and SMAP RSS version 4, L2 and L3 at two spatial resolutions: 40 km, hereafter referred to as RSS40, and 70 km, hereafter referred to as RSS70 for the same time period.

The JPL and RSS data products use different retrieval algorithms to generate SSS satellite data. Additional variations include corrections, flags, filters, masks as well as approaches to error and uncertainty estimation. The JPL product includes a land correction algorithm which uses a look up table and 'land-near climatology' to correct brightness temperature. JPL produces a 60 km product and its algorithm retrieves SSS within 35 km from the land wherever sea ice concentration values are less than 3% [27]. Version 4.0 of SMAP RSS mitigates land intrusion using simulated Aquarius and SMAP observations. RSS's method allows retrievals within 30–40 km from land. RSS produces both a 40 and 70 km SSS product; however, salinity retrievals degrade within 500 km of land [22]. The SSS 70 km product is often used as it is significantly less noisy than the 40 km data [22].

The JPL SSS product has been validated extensively and has shown good accuracy in validations carried out by other studies [28–30]. Tang [31] validated the SMAP JPL L3 SSS product using in situ data obtained from Argo floats, moored buoys and TSG. The results had a margin of error within 0.2 psu for the zonal band between 40°N and 40°S. Bao [13] investigated the accuracy of three L3 salinity products including the SMAP RSS and observed that SMAP RSS was positively biased in the tropics. Qin et al. [32] found in their evaluation of RSS SSS that undetected precipitation and strong winds biases SSS measurements. Fournier et al. [30] analyzed the performance of two satellite SSS products (including SMAP RSS) and found that close to the Amazon River RSS SSS products not only captured the full spatial extent of the plume during peak discharge consistently, but also the salinity gradients [30].

### 2.4. Hybrid Coordinate Ocean Model (HYCOM)

HYCOM is an assimilated model which, through the three-dimensional variational Navy Coupled Ocean Data Assimilation (NCODA) system, incorporates satellite observations as well as Argo float, CTD and Expendable Bathythermograph (XBT) measurements to forecast ocean variables such as salinity, currents and temperature among others [33]. The U.S. Navy uses HYCOM on a daily basis as part of their Global Ocean Forecasting System (GOFS) as does the National Oceanic and Atmospheric Administration (NOAA) at

the National Centers for Environmental Prediction (NCEP) [33]. The HYCOM data used in this study is from GOFS 3.1 and experiment 93.0, which has a 0.08° spatial resolution and a 3 h temporal resolution. In GOFS 3.1, SSS is initialised using climatology from the Generalized Digital Environmental Model (GDEM4). In-model relaxation to GDEM4 climatology occurs at locations where the model salinity values have a less than 0.5 psu difference from climatology salinity values. This is intended to improve model SSS calculations at river outflow areas [34].

NCODA runs on a regular basis at the Fleet Numerical Meteorology and Oceanography Center (FNMOC) as well as at the Naval Oceanographic Office (NAVOCEANO) and it supplies HYCOM with input parameters for SSS, SST and currents [19]. To do this, ocean observations are compiled and put through specific quality control methods. The quality-controlled data are then ingested by NCODA, which is run with data from the previous HYCOM forecast. The updated NCODA output is then used to initialize the final HYCOM simulation. HYCOM's fine resolution accurately resolves mesoscale ocean behavior and fast flowing western boundary currents [18]. These elements have made the model one of the favored choices for researchers to examine freshwater fluxes and salinity budgets [35] and to validate satellite salinity retrievals [14,21]. Due to the inability of Argo floats (one of the model's main sources of in situ data acquisition) to record measurements at areas close to land, HYCOM may have an issue recognizing freshwater discharge in these locations [21]. Overestimations of salinity by HYCOM in certain areas may also be due to deficiencies in the climatological forcing of the model, specifically in its ability to represent freshwater fluxes in its output [35,36].

### 2.5. Collocation and Validation Methodology

All (RSS70, RSS40, JPL and HYCOM) salinity datasets were collocated with saildrone observations. Data collocations were made using Xarray's interp method for multidimensional interpolation of variables [37]. The Xarray nearest-neighbour interpolation routine was used to match the locations and times of the saildrone product with the nearest location and time available in the averaged model and satellite salinity products. Only collocations within 24 h and 25 km were included, and the closest points in space were collocated before the closest points in time.

For the SMAP datasets, the L2 orbital data were collocated with the saildrone data using the Pyresample kd-tree resample_nearest method and SciPy spatial kd-tree method for quick nearest-neighbour lookup [38,39]. The multiple saildrone data points that matched with a unique SMAP observation were averaged to a single saildrone observation, providing a single collocation matchup. To compare the HYCOM model and SMAP data to the saildrone observations, the standard deviation of the difference (STD), the mean difference and the Spearman correlation coefficient between observed (saildrone) and each of the estimated (HYCOM, JPL, RSS40, and RSS70) salinity products were calculated. The L2 JPL and RSS datasets were preferred for statistical comparison to the saildrone data since these datasets have minimal spatial and temporal averaging as compared to the L3 datasets. Therefore, the analysis presented in the discussion uses the L2 datasets (Section 4).

## 3. Results

### 3.1. Accuracy of Saildrone Observations

The accuracy of salinity measurements by each of the three saildrones deployed was assessed by comparing individual salinity sensor (SBE37 and RBR) measurements on board each vehicle. For our analysis we use the SBE37 data as recommended by the ATOMIC Cruise Report [40]. Table 1 provides statistical analysis for the duplicate sensors-mean difference, standard deviation difference (STD) and root mean squared error (RSME). There is a consistent small offset (~0.06 psu) between the means of the SBE37 and RBR data on each vehicle that is attributed to a constant calibration offset. The standard deviations are ~0.002 psu, indicating that variability is consistently observed by both sensors. Additionally, the root mean squared error (RMSE) values for each vehicle are less than 0.07, which

indicates that the two sensors produced quite similar datasets. These results confirm the consistency and accuracy of saildrone salinity data across sensors highlighting that salinity data from calibrated saildrone salinity sensors are ideal for validating conventional remote sensing techniques in tropical coastal regions.

**Table 1.** Mean difference, standard deviation difference (STD) and root mean squared error (RMSE) between each saildrone's SBE and RBR instruments used to retrieve salinity from the western Tropical Atlantic, between 17 January–2 March 2020.

| Vehicle | SBE37-RBR Salinity (psu) | | | |
|---|---|---|---|---|
| | Mean | STD | RMSE | Number of Samples |
| 1026 | 0.052 | 0.002 | 0.052 | 66,240 |
| 1060 | 0.057 | 0.002 | 0.058 | 66,240 |
| 1061 | 0.059 | 0.0004 | 0.061 | 66,240 |

### 3.2. Comparison to Satellite and Model Salinity

Figure 4 shows the temporal plots of salinity for each saildrone versus the three satellite salinity datasets described earlier. The SMAP SSS values are represented as points and the saildrone and HYCOM values are represented as lines. This is to account for and represent the variations in time between the saildrone and SMAP measurements. L2 SMAP data from the JPL, RSS70 and RSS40 were able to capture the main salinity features detected by the saildrones. SSS from the saildrones showed 3 fresh tongues during the 45-day cruise. These are characterized by a 1.5 psu decrease in salinity on 5 February, a 1 psu decline on 7 February and the most significant decline of 2 psu during 16–19 February (this is discussed further in Section 3.4).

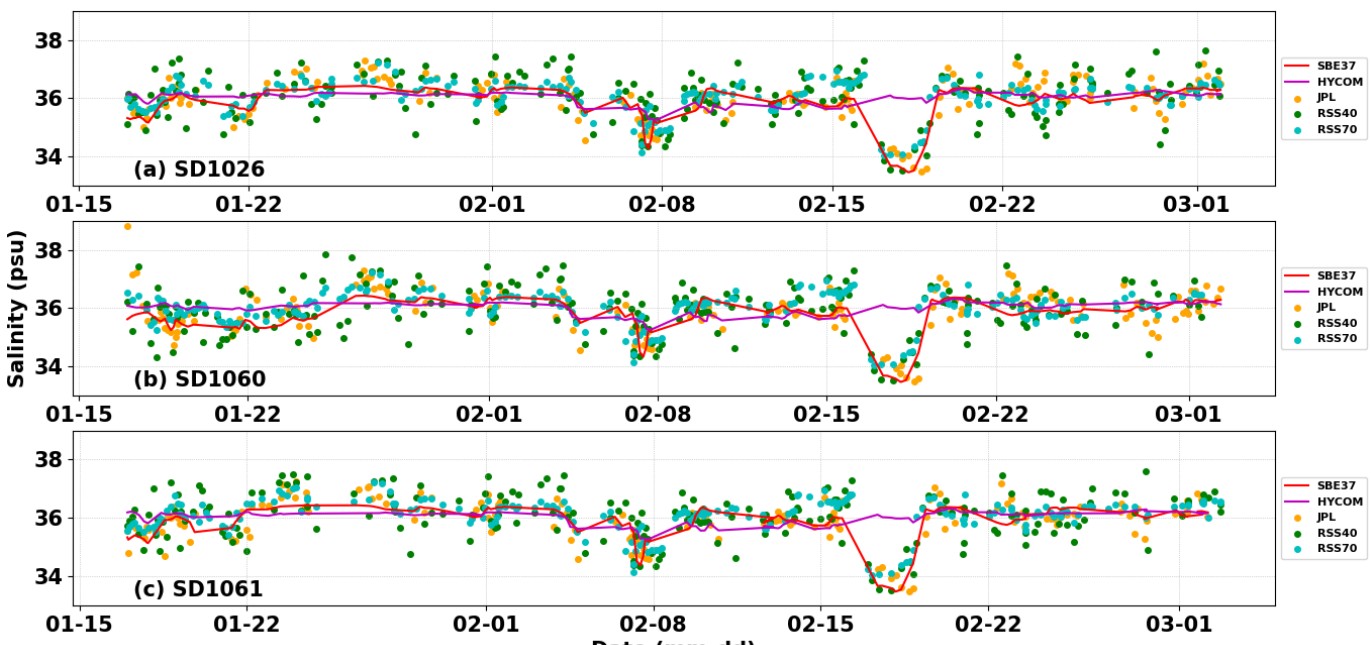

**Figure 4.** Salinity time series from SBE37, JPL, RSS40 and RSS70 for (**a**) saildrone 1026, (**b**) saildrone 1060 and (**c**) saildrone 1061 from 20 January–3 March 2020.

The saildrone and remotely sensed data were in general agreement, with differences that are not temporally dependent. Consistent with the earlier discussion, RSS40 displays the greatest variability. All satellite SSS clearly measure the strong fresh tongue, but the HYCOM model did not detect the fresh tongue. To further analyze the relative performance

of the SMAP and HYCOM salinity products, statistical comparisons along the entire saildrone track, for the duration of the cruise, are shown in Table 2.

**Table 2.** Mean difference and standard deviation difference (STD) for saildrone minus satellite and HYCOM data, as well as Spearman correlation coefficient from saildrone and satellite/HYCOM between 17 January and 2 March 2020.

| Product | | 8-Day L3 Salinity | | | | Orbital L2 Salinity | | | |
|---|---|---|---|---|---|---|---|---|---|
| | | Mean (psu) | STD (psu) | Spearman Correlation | Number of Samples | Mean (psu) | STD (psu) | Spearman Correlation | Number of Samples |
| 1026 | JPL | −0.166 | 0.241 | 0.859 | 208 | −0.113 | 0.460 | 0.631 | 160 |
| | RSS70 | −0.273 | 0.270 | 0.812 | 206 | −0.187 | 0.398 | 0.593 | 210 |
| | RSS40 | −0.279 | 0.323 | 0.759 | 206 | −0.188 | 0.658 | 0.432 | 210 |
| | HYCOM | −0.127 | 0.520 | 0.560 | 208 | −0.105 | 0.522 | 0.544 | 160 |
| 1060 | JPL | −0.173 | 0.257 | 0.835 | 222 | −0.105 | 0.545 | 0.584 | 169 |
| | RSS70 | −0.298 | 0.261 | 0.857 | 221 | −0.210 | 0.388 | 0.672 | 212 |
| | RSS40 | −0.298 | 0.312 | 0.791 | 221 | −0.159 | 0.666 | 0.481 | 212 |
| | HYCOM | −0.183 | 0.509 | 0.487 | 222 | −0.172 | 0.528 | 0.475 | 169 |
| 1061 | JPL | −0.158 | 0.246 | 0.868 | 196 | −0.012 | 0.449 | 0.662 | 160 |
| | RSS70 | −0.273 | 0.261 | 0.836 | 194 | −0.201 | 0.391 | 0.601 | 209 |
| | RSS40 | −0.271 | 0.320 | 0.771 | 194 | −0.198 | 0.664 | 0.411 | 209 |
| | HYCOM | −0.131 | 0.534 | 0.541 | 196 | −0.099 | 0.519 | 0.541 | 160 |

Table 2 provides the mean difference, standard deviation difference (STD) and the Spearman correlation coefficient for JPL, RSS40, RSS70 and HYCOM SSS data as compared to the SSS data from each saildrone. All repeat observations have been removed and observed salinity data is from the SBE37 instrument on each saildrone. For both the 8-day L3 Salinity and the Orbital L2 Salinity, the 3 h HYCOM product was collocated and interpolated to the times and locations of each of these datasets. These collocated HYCOM values are compared to their corresponding saildrone values to calculate the statistics displayed in Table 2.

For each of the L2 and the L3 salinity SMAP datasets, the saildrone observations were fresher than the satellites and model SSSs. The L3 data are consistently more saline than the L2 data. This could be because the L3 dataset consisted of an average of high salinity data points as compared to the individual L2 data points. Averaging in L3 products smooths a transient fresh tongue event that may be present in only 1 or 2 days of the 8-day average. However, in the individual data points in the L2 dataset, the fresh tongue would be more prominently represented.

For the L2 values, the RSS40 dataset has the highest STD values overall, HYCOM had the second highest STD values, followed by JPL and then RSS70. For the L3 STD values, HYCOM had the highest STD, followed by the RSS40, then RSS70 and then the JPL. The L3 STD values are smaller than the orbital L2 STD values, which is most likely due to the reduced noise in the L3 dataset because of the inclusion of averaged data.

The Spearman correlation coefficients for the L3 data were higher and therefore more closely correlated to the saildrone data than the orbital data, which suggests that the linear variability in saildrone data is closely related to the linear variability in L3 data versus orbital data. This is likely due to the reduced noise in the L3 dataset. For both datasets, the RSS70 showed moderate positive correlation with saildrone 1060, whereas the JPL showed the highest positive correlation to data from saildrones 1026 and 1061. For all three saildrones, HYCOM had the second lowest correlation coefficient and RSS40 had the lowest as it is a higher resolution and noisier dataset. The other variations in STD and Spearman coefficient values will be discussed further in Section 4.

Figure 5 explores salinity dependence in the satellite retrievals. RSS70 (Figure 5a) has the least spread of all the datasets above 35 psu indicating good agreement but appears to be too salty in the fresh tongue as reflected by the preponderance of delta-SSS values

less than 0. RSS40 (Figure 5c) data points are the most spread, which corresponds with the high noise content of the dataset. Additionally, all datasets appear to show a negative salinity bias which suggests that the model/satellite tends to overestimate salinity. The JPL appears to have the smallest negative bias out of the satellite datasets, and it was found that 56.8% of all JPL measurements for all three saildrones were overestimating SSS, as compared to 73% of RSS70 measurements and 61.6% of RSS40 measurements.

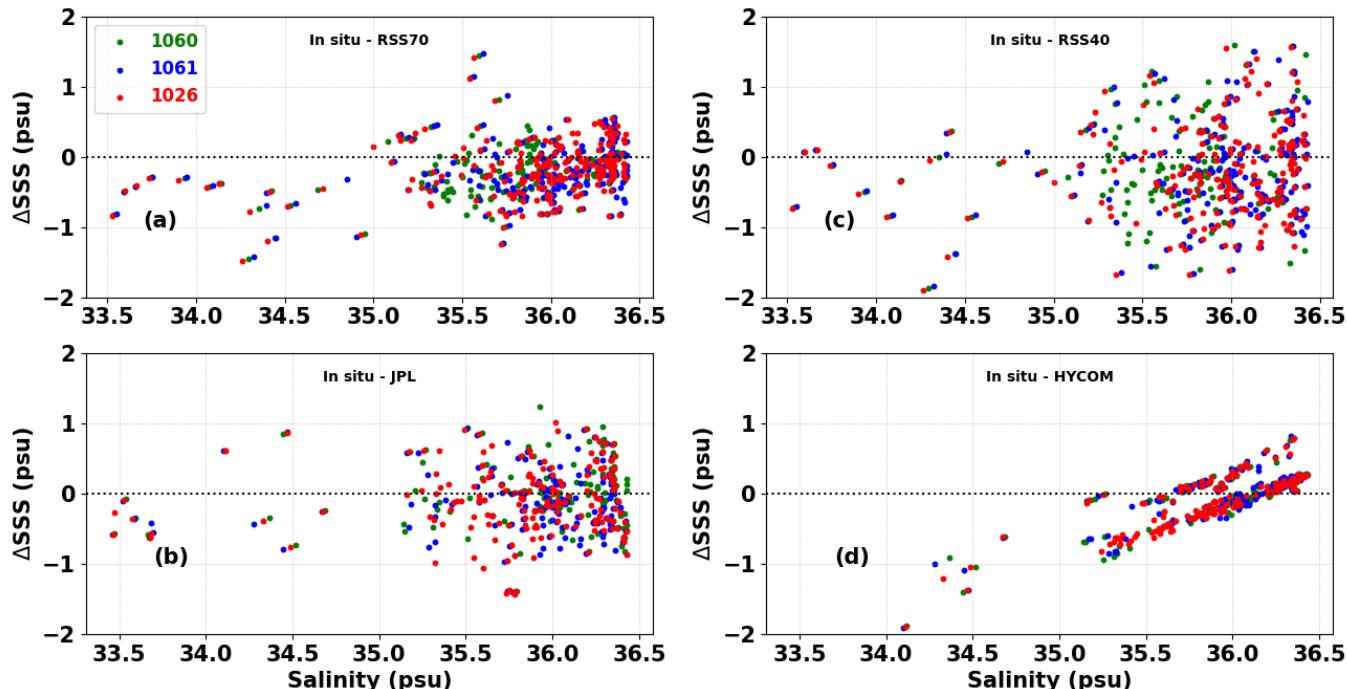

**Figure 5.** Sea surface salinity difference versus sea surface salinity between (**a**) RSS70, (**b**) JPL and (**c**) RSS40 (orbital L2 no repeat data) and (**d**) HYCOM and the saildrone instrument SBE37 as recorded by saildrones 1026, 1060 and 1061.

The HYCOM data (Figure 5d) is the least spread dataset of all, which is an indication of the lack of noise in the data due to the model's SSS relaxation The majority of the HYCOM SSS values were recorded within 1 psu difference of the saildrone SSS except for a few outliers which presumably were the values recorded within the fresh tongue. HYCOM SSS appeared to be at maximum 2 psu greater than the saildrone SSS in this area. The linear relationship between saildrone and HYCOM SSS implies that the model's SSS relaxation to GDEM4 climatology limits its salinity values to a certain range.

### 3.3. Spatial Visualization of Fresh Tongue by SMAP Products

Figure 6 shows the L3 salinity data centered on 17 February 2020 from the SMAP and HYCOM products in the background. Overlaid are the salinity measurements recorded by the saildrone between the 16–20 February 2020. The SMAP data in Figure 6 shows less variability than the saildrone measurements, due in part to differences in spatial and temporal resolution. The JPL product is smoothly varying with some potentially erroneous values in several pixels directly adjacent to South America that could be due to issues with the land mask correction or could also potentially reflect real coastal variability.

Near-land retrievals: The RSS product masks regions near land, and generally matches the JPL retrievals while also appearing to be marginally more saline. There appears to be a fresh 'halo' around the land in both RSS products that is not present in any other data. The RSS40 salinity has large biases near land that are not present in the JPL product as well as additional fluctuations, without any clear indication of improved spatial resolution. Near land, the JPL additional retrievals match the HYCOM data and appear to add value.

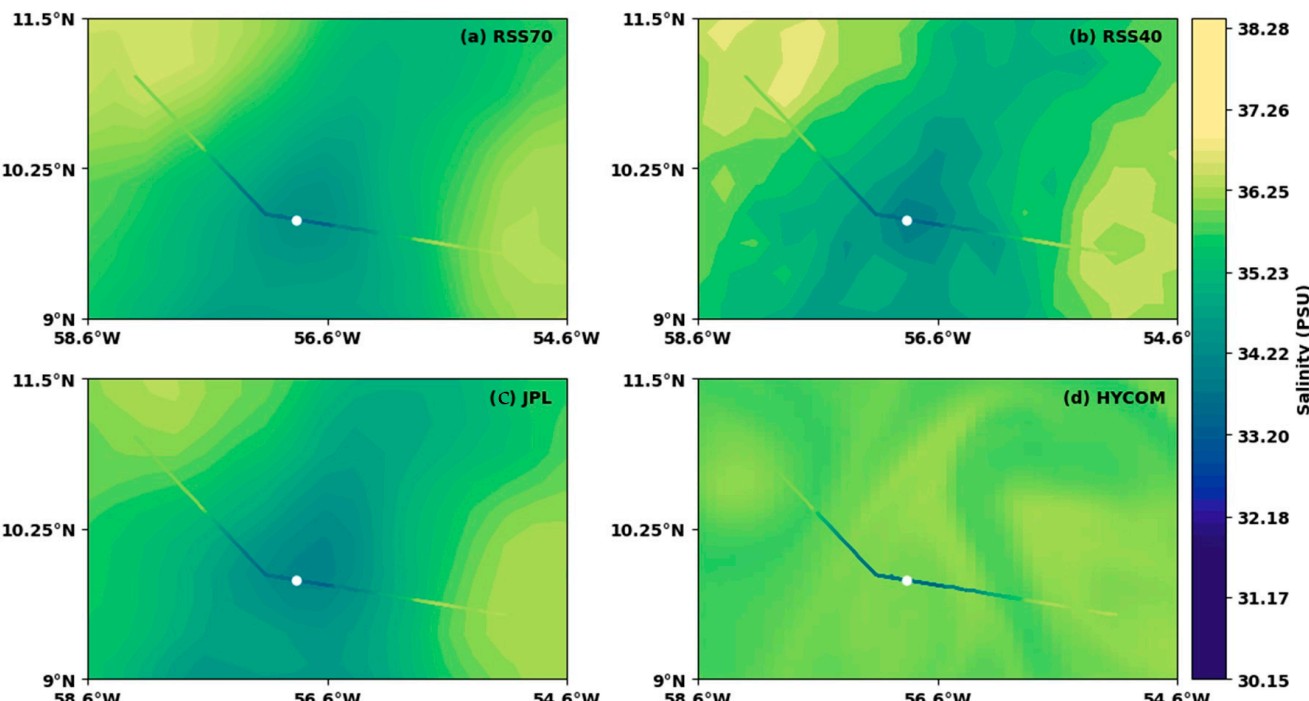

**Figure 6.** (**a**) Map of RSS70, (**b**) RSS40, (**c**) JPL and (**d**) HYCOM salinity for 8-day average centered on 17 February 2020 shown behind the 1026 saildrone vehicle's SBE37 salinity data from the 16 February to the 20 February 2020. On 18 February 2020, the saildrone crossed the fresh tongue, the vehicle position indicated by white dot.

Fresh Tongue: All versions of the SMAP satellite observations identified the fresh tongue but recorded moderately more saline waters across the fresh tongue when compared with the measurements of the saildrone, measuring a 33.33 psu minimum (18 February 2020 08:59:00) in contrast to the 33.49 psu, 34.03 psu and 33.52 psu of the JPL, RSS70 and RSS40, respectively. HYCOM fails to reproduce the fresh tongue event in its prediction as seen in Figure 6.

### 3.4. Investigation of Fresh Tongue

The three saildrone vehicle observations were very close to each other while within the fresh tongue (Figure 4). Hence, one of the vehicles (SD1026) was chosen for an analysis of the temperature and ocean conditions at and over the tongue. According to Reverdin et al. [5], this fresh tongue originated from a freshwater plume off the coast of South America. They also noted that the fresh tongue was transported to this area by mesoscale eddies as well as Ekman Transport. The saildrone salinity values recorded in this feature, observed in Figure 4, are relatively similar to the SMAP estimates and less saline than HYCOM estimates. These data provide an opportunity to better understand this feature and improve both the satellite and modeled products. Figure 7 compares the time series of the saildrone, SMAP JPL and RSS salinity data as the saildrone crossed the fresh tongue. Figure 7a puts the SSS into focus by comparing the salinity data observed by the saildrone with salinity data from RSS40 and RSS70, JPL and HYCOM. The figure shows that the satellite products were able to resolve the fresh tongue SSS, albeit not with as much detail as the saildrone data. Despite resolving small scale variability, the RSS40 was the only satellite product to capture the observed small increase in salinity at approximately midday on 18 February 2020. As previously stated, HYCOM does not simulate the fresh tongue event.

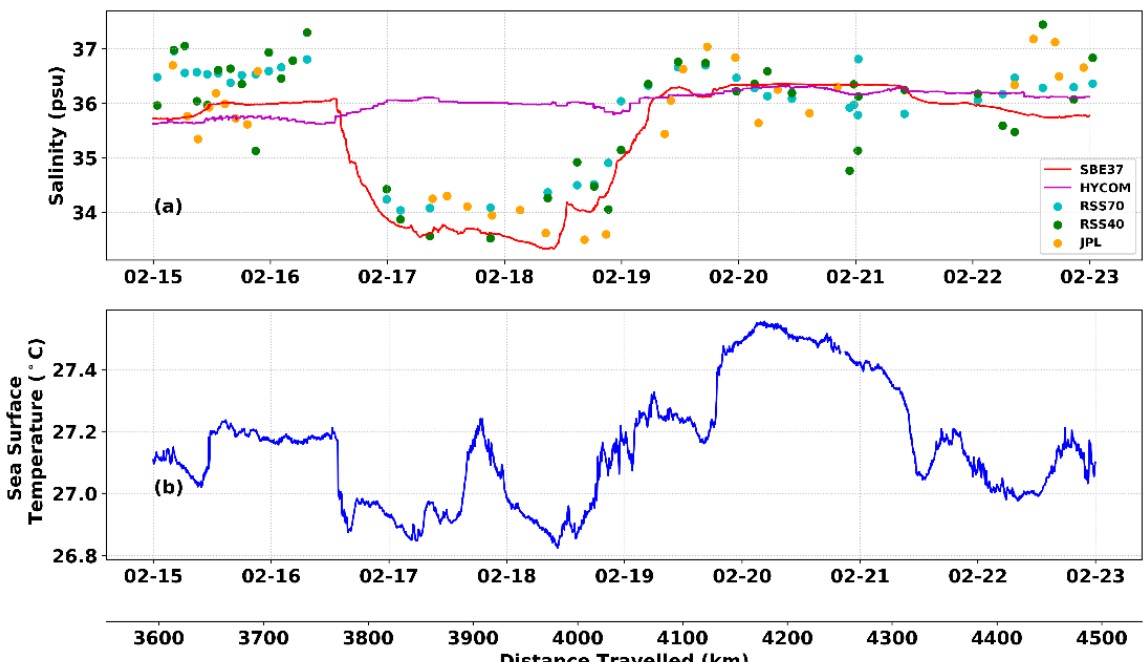

**Figure 7.** Time series of (**a**) the salinity measured by the SBE37 sensor onboard the saildrone 1026 compared with that of HYCOM, RSS70, RSS40 and JPL satellite salinity products and (**b**) the sea surface temperature measured while saildrone 1026 crossed the fresh tongue southeast of Barbados from 16 February 2020 12 UTC to 19 February 2020 9 UTC.

Upon entering the fresh tongue, the SST fell by approximately 0.3 °C and the SSS also experienced a reduction of approximately 2 psu (Figure 7a,b). This decline in SST and SSS reveals that the saildrone crossed a cold core eddy.

## 4. Discussion

The EUREC$^4$A-OA/ATOMIC campaign highlighted the effectiveness of the saildrones in measuring SSS across the tropical North Atlantic Ocean. Saildrones were directed to areas of significant interest in the NBC region and were able to provide exceptionally high spatial and temporal resolution data. The high quality of the data was reflected in the low RMSE values produced during the comparison of duplicate sensors on each saildrone (Table 1). The saildrones also augmented the near surface data collected by the four research vessels that took part in the campaign as they generally start observations about 5–7 m above and below the sea surface [41]. A few months after the campaign, the shutdowns associated with the COVID-19 pandemic restricted field campaigns including ship-based research activities. As a result, remotely piloted or autonomous data collection platforms, such as saildrone, have become increasingly valuable given their remote operation and their ability to continuously collect in situ data over periods extending from months to years. The aforementioned benefits of the saildrones made them ideal candidates for collecting data to validate the satellite-derived marine products.

In this study, SSS from three saildrone vehicles were used to assess the quality of satellite and ocean model data. The differences seen in Table 2 show that the RSS70 and JPL estimates were more closely correlated to the saildrone data than equivalent HYCOM and RSS40 products and the STD values indicate that the RSS70 SSS may have been most accurate to the saildrone data [22]. The RSS40 data was the noisiest out of all four platforms and this high noise content was represented by its STD values and Spearman correlation values. This noise is most likely due to reduced spatial smoothing. However, some of this noise could be groundwater discharges and other land-based discharges from small river systems which are not visible in the lower resolution RSS70 and JPL output.

These results suggest that for most applications, the RSS70 data should be preferred to RSS40, which is in agreement with previous validation papers [20,42]. There also appears to be a correlation between the spatial resolution of the SMAP satellite and the precision of its salinity estimates. It was found that the satellites with coarser resolutions (JPL at 60 km resolution and RSS70 at 70 km resolution) and less noise resulted in more accurate output. Additionally, the mean difference values from all saildrones minus satellite/model values showed that SMAP and HYCOM overall tended to record more saline conditions than the saildrones.

The most prominent feature to occur over the course of the saildrone cruise was the salinity minimum experienced during 16–19 February 2020. This sharp drop in salinity was spatially visualized in Figure 6 to examine how well the SMAP products reproduce the saildrone data. The fresh tongue was adequately displayed by the JPL and RSS products, showcasing the products' ability to resolve fresh mesoscale features. The RSS40 produced the most accurate salinity data inside the fresh tongue. It was the only product that was able to depict the sharpest rise in salinity inside the fresh tongue, which is presumably because it has the highest spatial resolution of the satellite datasets. This lines up with Meissner et al. [22], where one of the suggested applications of RSS40 are areas of freshening events of the surface layer of the ocean. Thus, RSS40 may be suitable for investigating small scale salinity anomalies, but RSS70 should be the preferred datasets for most ocean salinity research and investigation in the western Tropical Atlantic [22]. Due to the in-model climatological relaxation which results in smooth data, HYCOM overall appeared to have the least noisy dataset. It also had several measurements in good agreement with the saildrone, but it failed to reproduce the fresh tongue. This is most likely due to an underestimation of the freshwater flux entering the model domain from the Amazon River system. Since the main sources of HYCOM's assimilated salinity data are not deployed close to land, freshwater discharges from rivers would be underestimated in model. River data, which accounts for freshwater runoff from the Amazon and other river inputs along the South American coastline, are required to improve model performance in the area of the fresh tongue as demonstrated by Coles et al. [43].

In the Coles et al. [43] study, HYCOM was initialized using surface forcing from the European Centre for Medium-Range Weather Forecasts (ECMWF) 40-year analysis (1979–1998) and a salinity restoration condition to prevent the model from using its default seasonal salinity cycle. Climatological discharge information from 315 rivers was included as climatological mass flux input and a 17 term empirical equation calculated the near field plume dynamics. In their HYCOM output, Coles et al. relied on the 35 isohaline to identify the Amazon plume. The salinity within their plume remained under 35 psu as the plume moved into the open ocean off the coasts of South America. This is not consistent with the HYCOM SSS variable as Figure 7 shows the HYCOM variable consistently above 35 psu, indicating the need for specific river input data to produce more accurate river outflow salinity estimates.

SMAP SSS data, which more accurately represented the fresh tongue in this instance, could potentially be incorporated into HYCOM's data assimilation process to have a more accurate representation of freshwater fluxes in the model. Such an approach has not been described in the literature to our knowledge and represents a promising new area of research that could improve the prediction performance of HYCOM.

## 5. Conclusions

During the EUREC[4]A-OA/ATOMIC campaign, saildrones and research vessels were deployed in the North Brazil Current region in order to investigate mesoscale eddies and submesoscale fronts. In this study, saildrone salinity observations recorded during the campaign were compared with four different SSS products SMAP JPL, SMAP RSS 40 km, SMAP RSS 70 km and HYCOM, with the aim of highlighting the strengths and weaknesses of each product and identifying the best product for use in this area. This study represents the first validation of SMAP satellite-derived SSS using the saildrone in

the river-influenced Western Tropical Atlantic, and this information can allow researchers to make informed decisions regarding the most ideal product for their application as well as highlight issues to algorithm developers. Overall, it was found that SMAP RSS 70 km outperformed its counterparts. However, SMAP RSS 40 km was better within freshening events such as a fresh tongue, with HYCOM being on the opposite spectrum, failing to identify submesocale features such as the fresh tongue. Akin to the SMAP RSS 70 km, SMAP JPL and HYCOM performed well in areas where the large-scale salinity conditions were constant, but improved input SSS fields may be required for HYCOM to reproduce smaller-scale salinity fluctuations. Finally, the results of this study can aid in the improvement of mesoscale and submesoscale SSS products, which can lead to the refinement of NWP and climate models and therefore to improved weather, ocean and climate forecasts, especially for tropical regions. An increase in SSS observations (e.g., augmented Argo floats or buoys) in the western Tropical Atlantic would allow for improved validation of SMAP and model estimates in this area. The use of the three saildrones for such a short period is insufficient to ascertain all of the SSS complexities in this location.

**Author Contributions:** This research paper was conceptualized by C.L.G. Software was developed by K.H., A.D., S.W., S.S. and C.L.G. The validation process was executed by S.S. The formal analysis was done by K.H., A.D., S.W., S.S. and C.L.G. Resources were provided by the Caribbean Institute for Meteorology and Hydrology and Farallon Institute and the data was curated by C.L.G. Writing-original draft preparation was executed by K.H., A.D., S.W., S.S. and C.L.G. and review and editing included K.H., A.D., S.W., S.S. and C.L.G. Data visualization was executed by K.H., A.D., S.W., S.S. Figure 4 was created by S.W., Figure 5 was created by S.S., Figure 7 was created by A.D. and Figures 3 and 6 were created by K.H. The project was supervised by C.L.G. All authors have read and agreed to the published version of the manuscript.

**Funding:** This research funding was provided by NASA Physical Oceanography grant number 80NSSC20K1003. EUREC$^4$A is funded with support of the European Research Council (ERC), the Max Planck Society (MPG), the German Research Foundation (DFG), the German Meteorological Weather Service (DWD) and the German Aerospace Center (DLR). Funding for the development of HYCOM has been provided by the National Ocean Partnership Program and the Office of Naval Research. Data assimilative products using HYCOM are funded by the U.S. Navy.

**Data Availability Statement:** The saildrone v1 data presented in this study are openly available at https://doi.org/10.5067/SDRON-ATOM0 (accessed on 30 April 2021). The HYCOM GOFS 3.1 data presented in this study are openly available at https://www.hycom.org/dataserver/gofs-3pt1/analysis (accessed on 17 September 2021), dataset id: GLBy0.08-expt_93.0-ts3z. The SMAP data presented in the study are openly available: JPL 8-day v5.0 https://doi.org/10.5067/SMP50-3TPCS (accessed on 30 April 2021), RSS 8-day v4.0 https://doi.org/10.5067/SMP40-3SPCS (accessed on 30 April 2021), JPL L2B v5.0 https://doi.org/10.5067/SMP50-2TOCS (accessed on 30 April 2021) and RSS L2 v4.0 https://doi.org/10.5067/SMP40-2SOCS (accessed on 30 April 2021). Software and data needed to reproduce tables and figures are available at https://github.com/cgentemann/paper_software/tree/master/2020_ATOMIC_Salinity (accessed on 13 May 2021).

**Acknowledgments:** The authors would like to acknowledge and thank David Farrell of the Caribbean Institute for Meteorology and Hydrology for providing the opportunities to participate in the EUREC$^4$A-OA/ATOMIC campaign and enabling collaborations with Chelle Gentemann of the Farallon Institute. Sarah Ann Thompson added references and proofread the document.

**Conflicts of Interest:** The authors declare no conflict of interest. The funders had no role in the design of the study; in the collection, analyses or interpretation of data; in the writing of the manuscript, or in the decision to publish the results.

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
