# Peer review of "Validating Salinity from SMAP and HYCOM Data with Saildrone Data during EUREC4A-OA/ATOMIC"

_remotesensing, doi:10.3390/rs14143375_

Round 1
Reviewer 1 Report
This is my third review of this paper. It still has a lot of issues as can be seen by the volume of comments below. Mostly it is a matter of better explanation of certain items.
The authors need to go back over the figure captions very carefully. Many of the comments below refer to inaccuracies in the captions. Captions can feel like an afterthought, but they are anything but. Many people encountering this paper after it is published will only look at the figures and read the captions!
As noted below, the collocation method described in section 2.5 is still murky. I would also note that the same confusion has come up in previous reviews. The authors need to think carefully about how the matchups were done and how to describe it. This is essential for reproducibility.
44. "North" or "tropical Atlantic..."
53-54. "Several studies" implies more than one or even two. I have not read reference [3] in detail, but from the title and abstract, it does not seem like the correct one.
64. "poorly recorded" I would use the word "sparse" instead.
106. "are". "have been"? "will be"? Is this a statement of what will be covered in the paper, or a reference to past work? If the latter a reference needs to be provided.
Figure 2. The BdT term is still in here, despite my having asked for it to be removed in three previous reviews! They even said "removed" in their most recent response, but it remains. Can the authors please explain why they are refusing to do this.
175. "Profiler"
Figure 3. The blue color indicates...
203-204. "Hence, it is important to independently assess the accuracy of the data..." The next paragraph (lines 205-210) gives a couple of instances where this was done in other field campaigns (the SPURS-2 example needs a reference), but does not say anything about how the saildrone data were quality-assessed during ATOMIC.
210-215. These are examples of satellite data being validated using saildrones, not the validation of saildrone data themselves.
216. "The Soil Moisture Active Passive (SMAP)" Alternate wording "SMAP data"
220. SMAP does not detect soil moisture. It detects brightness temperature, which can be converted to soil moisture on land. "SMAP is primarily designed to measure soil moisture, but also measures ocean surface salinity..."
228. "offer" Better word: "use"
237. "500 km of land" Add a reference to [22]
237. In fact the 70 km product is what RSS recommends using in most applications. See reference [22].
273. What is an "over-prediction"?
274. "climatological forcing" I don't work with HYCOM in detail... My guess is that it is forced with near-real time winds and surface fluxes, not climatology. The authors can correct me.
277-279. These two sentences are repetitive, but contradictory. Previously (lines 224-226) the authors said they are using L2 as well as L3 SMAP data, but here seem to imply they are using only L3. The L3 data are provided on a 1/4°grid (a fact unmentioned in section 2.3). So a collocation distance between a saildrone observation and a SMAP one could be as large as 4 days and 12.5 km. Why did the authors use L3 data for validation instead of the more specifically located L2 data? (Or maybe they did - lines 288-290).
279. It is not clear what the word "prioritized" means in this context.
294-295. The stuff in the parentheses is unnecessary. Ditto lines 336-337.
301. This (0.0012) is not the number given in Table 1.
325. "a small temporal delay" I cannot see what this is referring to. Which of the three events?
337. "averaged" I am trying to figure out what the authors mean by this. No averaging is discussed in section 2.2, except maybe for 1-minute saildrone values. But the data displayed in Figure 4 do not look like 1-minute averages. I would also note that the HYCOM values discussed a couple of lines further on are not averaged, but collocated and interpolated.
341. "column" I think the authors mean "row".
Table 2. For esthetics I would recommend putting a vertical line to the left of the second "Mean" column.
342-347. I cannot understand this reasoning about the mean differences between saildrone and satellite/model values. There is something about a cold tongue, but I don't know what that has to do with SSS. Can the authors try to articulate this better, or think more deeply about what the issue might be.
368-369. "Each panel...HYCOM" Repetitive from the figure caption. Delete.
378-382. The authors need to comment on the obviously different characteristics of the differences in HYCOM relative to the satellite datasets, i.e. the systematically linear change going from low to high. There is something fundamentally different about HYCOM, not just that it is less noisy.
Figure 6. Most of this figure is not discussed, only what is in the red box. The authors should reduce the size and scope of the figure to focus on that box and better make their points with regards to the fresh tongue of the Amazon.
417. "three saildrone vehicles" (Add an apostrophe!) I only see one track. Were the saildrone tracks and SSS values so close as to be indistinguishable? This is not clear from Figures 3 or 4. Maybe this is only one saildrone? I would recommend only putting one saildrone on here if they are really that close. And by the way, this figure is much improved from the previous version - at least what I remember of it. OK, I guess it is only one saildrone (line 422). The caption needs to be updated.
420 and 431. Why two separate sections? And section 3.5 includes analysis of the fresh tongue by satellite and model, contrary to the title of the section.
Figure 7c is not discussed and can be deleted. Ditto for the air temperature record in Figure 7b. The caption is incorrect and needs to be updated.
471-473. I do not see what this is talking about. There is an unexplained gap in the JPL (and all the other satellite/model) dataset as the saildrone entered the fresh tongue. If the authors think this is real, they should give an estimate of the time lag, and come up with a better explanation - or better explain their hypothesis.
475. The authors should cite Meissner et al's report [22].
480-484. I think this sentence is not gramatically correct.
496-497. Again a citation to the Meissner et al report would be appropriate.
507. "their study" Whose?
513-514. This is circular reasoning. The plume was defined as being less than 35, so it remained less than 35.
514-517. I looked in section 2.4 for information about exactly how HYCOM incorporates river input, but could not find anything useful. Since this is a crucial part of the analysis of the paper, the authors need to discuss this at length and in detail. I have not looked at the Coles et al paper that is cited here, but that looks to be a one-off test of the model, and not the standard product that is used in this paper.
591. I'm not sure I would call this a "halo". It seems to be one very fresh pixel, and it is located very close to the mouth of the Orinoco river. I think the authors are over-interpreting this one observation and an adjacent one to the southwest. Maybe they have some reason to connect this with the brightness temperature correction other than pure speculation. That said, of course, the conclusion in lines 523 and 524 is obvious.
542. "irregularities" This is a strange term, used in this section and in the abstract, but nowhere else in the paper. The authors should either define it and discuss what it means in the text, or use a word that is more familiar to their audience.
545. "SMAP JPL
582. This URL was broken in the process of converting to pdf. The authors should make sure this does not happen in the proofing process.
591-595. This information belongs in the "Funding" section.
Author Response
Please see attached doc.

Reviewer 2 Report
I am happy with the adjustments made in the manuscript.
Author Response
Reviewer happy with paper, no comments returned.
Reviewer 3 Report
The authors have successfully addressed my questions. The paper is ready to publish.
Author Response

(The authors gave the same response as above.)

Round 2
Reviewer 1 Report
This is my fifth (and last) review of this paper. I mainly went through to see if my previous comments have been responded to reasonably.
56. The studies cited here, Bao et al., Tang et al., and Qin et al., do not show any large bias specific to the tropical Atlantic that I can see, especially any based on in situ platforms. And I am not sure what the word "persistent" refers to. Is the bias high or low?
174. I think [] numbered citations are used in RS. The editor should clarify.
Figure 2. I still think the quality of this figure is poor. It's not so much the clouds, but the maps, arrows, and fonts that are pixellated. I realize this is taken from another paper. Maybe the original authors have a higher resolution version they can contribute.
195. "blue color indicates the ocean". Maybe the blue shading is the depth? What doe the variations in the shading mean? It would much more useful to put real depth information in this figure so the reader can tell how far the continental shelf extends. Some of this is in Figure 1, but a more detailed version of the inset of Figure 1 would be great.
207. "Hence, it is important to independently assess the accuracy of the data..." I guess this is what Section 3.1 is for. Though it seems like this is not an assessment of the saildrone data against some other platform, but a comparison of two different sensors on the same saildrone. Is this the assessment that is being referred to here? If not, then the authors should add a reference to a different study. Honestly, since the data are being collected by a Seabird CTD, the only real assessment needed is a pre- and post-cruise calibration.
220. "Soil Moisture Active Passive (SMAP data)" This is awkward. Just use "Soil Moisture Active Passive data" for this heading.
264-268. In their response, the authors state that "The version of HYCOM used in this paper does not appear to incorporate river input directly into the model." This statement should be in here somewhere as it says clearly what the issue with HYCOM is in this region. And the authors need to verify that this is true and remove the words "appear to".
355-356. Maybe take out "when compared to saildrone". As it stands this is some kind of three-way comparison which is not clear.
Figure 6. The authors have opted not to reduce the scope of this figure as I recommended. (There is a much better version of the figure in the review response, but not in the manuscript.) I still think it would be much more informative if reduced in scope to the size to the red box, or somewhere close to it. It nicely illuminates the relationship of the saildrone SSS to the position of the fresh tongue in the different products.
Also Figure 6. I see 8 panels, two identical sets of 4. I presume this is part of the editing process and one of the duplicate sets will be removed.
Author Response
Please see the attachment.

This manuscript is a resubmission of an earlier submission. The following is a list of the peer review reports and author responses from that submission.
Round 1
Reviewer 1 Report
Hall et al: Validating salinity from SMAP with saildrones and research 2 vessel data during EUREC4A-OA/ATOMIC
This paper validates and compares SSS satellite products and an ocean forecasting numerical model, against in situ measurements collected during a scientific cruise in the region of the North Brazil Current rings. The results are fairly clear, but the presentation appears very unfinished. In particular, repeatedly throughout the paper, results from a topic are presented before the topic is introduced.
Abstract
Fine, except for me it is strange to introduce abbreviations in the abstract in stead of the Introduction.
Introduction
Does not follow the standard order of information. Standard: start with overall problem, why it is important, what has been done earlier, knowledge gaps, and aims and objectives of the present study. I suggest approximately this: move lines 75-82 first, then lines 50-74, and end with information from line 31-46 + 83-88.
Line 45: abbreviation NASA SMAP is introduced in line 66…
Figure 1: would be good to include tracs/positions of Saildrone, TSG and CTD casts.
Materials and methods
Here there is loads of information about validations done in earlier studies, making it hard to understand what the present study is doing.
Line 91 Manufacturer of Saildrones is introduced, with reference in line 115. I should go here.
Line 194: ‘two (2)’ – why put in the number here?
Line 341: ‘three (3)’ – same as above
Line 338: at this point it has become very unclear to me that section 2.4 is about the present campaign, and that the ‘study’ (line 343) is actually the present study.
Results
Here I really miss a map of the tracks (see Figure 1). It could well be an additional figure to Figure 1.
Line 382: the highest standard deviation I see in Table 1 is 0.002 psu, not 0.01.
Line 385-387: this information should have been in Introduction
Figure 2: it is hard to see color differences between blue and green. Also, there appars to be a mistake in timing at the end of the time series for some of the data. Especially visible for SST from RSS?
Line 404-405: this information should be given when first introducing Figure 2.
Line 406-410: this information could also well be in Introduction
Line 430: here Table 2 is introduced after results in it were presented in line 426.
Line 434: the word ‘correlates’ has a specific statistical definition, perhaps you should use ‘corresponds’ in stead.
Line 443: your interpretation of ‘negative’ measurement differences is opposite of what is defined in Table 2. Table 2 says saildrone minus satellite, implying negative values mean higher satellite values.
Line 448: meaning of ‘strongest relationship’ – least error?
Line 452: use of word ‘correlates’, see comment line 434
Line 476: First sentence here should be included when introducing Figure 4 in line 465
Line 478: Here you are in line with Table 2; satellite data ‘slightly saltier’ than saildrone, in contrast with comment to line 443.
Line 504: French Guyana is not included on the map, Figure 1.
Line 507: meaning of ‘underestimated’? lower fresh water content?
Line 527-531: this is a tiny detail in the time series. Not sure you can interpret it like this.
Line 535-537: The currents here are not directly linked to the wind. If they were, they would have been weaker, more like 3% in stead of 10% of the wind.
Discussion
Line 578: the satellites overestimates, not ‘underestimates’ as you write (see comments to lin 434 and 478).
Line 582: a spatial resolution of ‘60km’ is higher (or use ‘finer’) than ‘70km’ and less (or use ‘coarser’) than ‘40km’. You write opposite.
Line 593-594: what is ‘13m’ and ‘200m’. Are they distances in meter or perhaps time?
Line 611: if HYCOM is using climatological forcing, you should not expect it to be able to model correct timing of and amount of river run off.
Conclusion
Line 617: I think this study’s contribution to understanding ‘their role in ocean-atmosphere interactions’ has been tiny. Perhaps not to be mentioned.
Author Response
Thank you, our comments are attached.

Reviewer 2 Report
This paper purports to compare saildrone and shipboard SSS data with satellite data in the North Brazil Current during the EUREC4A/ATOMIC field campaign. Such comparisons have been done successfully in the past (e.g. Vazquez-Curevo et al., 2019, doi:10.3390/rs11171964) and could be a valuable contribution in this region influenced by the NBC and its rings, the Amazon outflow and the North Equatorial Current. However...
I stopped reading on line 365 and have no comments on any of the paper's results. As is clear from the comments below, section 2 is very poorly crafted. Without a clear idea of what data and methods the authors are using, like collocation and interpolation methods, and specific datasets, there is little point in examining the results.
I am on the edge about recommending complete rejection or just major revisions. I am willing to give reading this paper one more try if the authors make major improvements, as I think this type of comparison is important. If they decide to resubmit, they should also do the same critical reading of their results and discussion sections (which I did not read) to make sure they are of high quality.
36. The distinction between these regions is not apparent on the map of Fig. 1.
37. "Boulevard des Tourbillons" Where does this term come from? Can this reference just be removed, everything between the commas? It is very poetic and descriptive, but is it necessary?
39 & 42. "... northwestward..."
45. Define "SMAP".
52. Of these three references, only #6 talks about the "BdT".
64. "were not obtained..." SMOS began providing such measurements a year earlier, in 2010. I suggest the authors read Reul et al (2020) to learn some of the history and cite it somewhere in here.
https://doi.org/10.1016/j.rse.2020.111769
66. "instruments such as..." To date, the only follow-on to Aquarius is SMAP. And SMAP is not an instrument, it is a mission.
72. "eliminate the limitations..." Sort of, but they have limitations of their own.
82-83. This assertion, that the Amazon outflow is important to the predictability of El Nino events, needs a reference. The ones provided on the previous line (16 and 17) do not work.
86. "HYCOM" This comes out of the blue. The authors need to motivate why it is important to validate this model.
87. "platforms" Saildrones, or the satellites and models?
104. "the EUREC4A-OA/ATOMIC cruise" I am not familiar with how E/A went. Does this refer to a research vessel cruise, or a saildrone deployment or something else? There is only one cruise referred to, but apparently multiple saildrones. Is each saildrone deployment a "cruise"? Ditto line 189.
118-122. This is general information about how saildrones are calibrated. What is needed is what was done to calibrate the specific sensors on the saildrones deployed during E/A.
123. "the saildrones" The same ones?
135. "Institution"
159. "46086" Where was this?
Section 2.1. I came out of this section without such basic information as how many salidrones were deployed, for what time periods, what tracks they took and what depth (height) the in-water (above-water) instruments were measuring at.
192. "8-day running means" This refers to an L3 dataset, whereas the lines above indicate that L2 data are being used exclusively. L2 data are not 8-day running means. They are instantaneous snapshots. What data are the authors using exactly?
193. "collocated" The details of this collocation matter as satellite L2 and saildrone measurements are almost never simultaneous. What kinds of space-time windows are the authors using? How close in space and time are the satellite and saildrone observations? Ditto line 355. The two paragraphs starting at line 355 purport to speak to this, but as described below, are not useful.
193. "repeated data" I cannot understand what the authors mean by this. The L2 data are not repeated, or maybe they are referring to saildrone data.
194-200. There should be references sprinkled throughout this paragraph where readers can go for more information about the separate algorithms for the two products.
209-239. While all of this is interesting, I am not sure what it has to do with the focus of the paper, i.e. comparison between satellites and saildrones and TSGs (see paper title). Maybe this will become apparent later. Ditto for much of section 2.2.2.
241. I think this is referring to this paper.
https://doi.org/10.3390/rs13010110
341 and following. OK, so now we learn there were 3 cruises on R/Vs. We are given the name of one of them (the L'Atlante), but not the other two. What were the tracks of these cruises, and over what time periods? We are given the specs of the TSG in the L'Atlante, but not for the two other unidentified vessels.
Section 2.4 and 354-364. No collocation method is given for the TSG data. The TSGs were sampled every 30 seconds (line 345). Did the authors really use 30-second TSG data to do comparisons with 8-day average satellite data? How?
357-360. These are very cryptic descriptions of the interpolation/matchup methods. Instead of referring to obscure python documentation (see comment on lines 747-752), the authors should explain how the methods work.
360. "is included" as an ancillary variable.
361. Yes, you already said this (168-169).
361-362. The authors are apparently matching 8-day 60-km averages with 1-minute point samples. There are many ways to do this matching, but which one they use is not clear.
Section 2. As indicated by the volume of comments above, this section is poorly crafted and needs a lot of work. There is stuff in here that is superfluous and can be removed. There are important details that are missing. The authors need to focus on what data were collected, and how the comparisons were done. They should delete everything that does not speak to this, and re-read this section with a critical eye and a fat red pen. What important information does the reader need to know? Anything that does not fit that description should be cut. Anything that is missing should be added.
747-752. These appear to be incomplete references. Are there DOIs? A google search did not come up with anything useful.
Author Response
Thank you, our comments are attached.

Reviewer 3 Report
Please see attached

Author Response
Thank you, our comments are attached.

Round 2
Reviewer 2 Report
This is my second review of this paper. The authors have much improved the intro and "Materials and Methods" section (see comment on line 109). However, there are still many issues they need to address before this paper becomes suitable for publication. See below for detailed comments. The main issue is that the authors need to better articulate the value or use of their study (lines 24-26). There is analysis of TSG data stuck onto the end of the paper that is out of place. Finally, the referencing and data availability statements have a lot of problems.
The inclusion of HYCOM in this paper highlights the fact that this model is problematic, especially in coastal regions and ones with large variability. It is commonly used for validation of satellite data (e.g. Meissner et al., 2018). Its use for this purpose should be reconsidered carefully in light of what the authors have shown.
The citation on the first page has the wrong title.
22. "with" Do the authors mean "without"?
24-26. These are excellent goals and motivations for such a study. However, I cannot see in the discussion section where any suggestions are made to this effect. How do the authors think the results of their work can be used to influence and improve the different satellite products or HYCOM?
51-52. Again, this terminology "BdT" comes out of nowhere. What is the BdT? I looked in references 8 and 10 for the definition with no success. See also comment on lines 626-627. I guess there is a box in Figure 1. But why give it this odd name instead of just calling it a box in the western NA? Is it the same as the North Brazil Current region (line 93)?
65. SMOS did not start transmitting data until 2010.
67 and 69. Why is the same information not given for SMOS?
75-85. This is an elementary introduction to ocean modeling that might also be found in a textbook. However it does not motivate the effort to validate HYCOM in any useful way. The authors mention data assimilation. Does HYCOM assimilate saildrone data? What are the authors trying to learn about the model? How can what they learn be used to improve it?
82. "climatological" I would remove this word.
109. Suggest calling this section "Data & Methods".
119. What is the "EUREC4A-Circle"?
133. "additional" In addition to what?
159. I thought saildrones could only carry one additional sensor on the keel (line 147).
165. "a cruise" This seems to refer to a single cruise, but there were 3 saildrones. Is this single cruise a reference to the collective missions of the 3 different saildrones. Ditto line 250.
Figure 2. 1) There is an inset map of saildrone tracks, but it is not clear where the inset would be placed within the figure. 2) The quality of the figure is very poor. The graphic is pixellated. 3) What is "poldirad"?
196. RMSD between the saildrone and what?
280-283. This refers to a set of python routines. I guess one could look up what these routines do in the documentation. However, this is not a useful description of how the matchups were carried out. Apparently the collocations are done within 24 hours and 25 km, but the authors do not say anything about how they balance space and time in their (I guess) nearest neighbor scheme. This kind of detail is important as spatial differences likely matter much more than temporal ones at this scale.
289. What is an "orbital satellite dataset"?
297. Repetitive from lines 161-162.
338-340. "may have a spatial mismatch". What does this mean? What are the authors trying to say in this sentence?
Table 2. HYCOM has listed values of mean, STD, etc. for both the "8-day L3 Salinity" and the "Orbital L2 Salinity" columns. What is the difference? Maybe this is stated somewhere in the text.
357-359. This sentence needs rewriting. It's not clear what is being correlated to what?
396. Given the color scale of Figure 5, HYCOM displays very little variability at all, or any correspondence to the saildrones.
397-404. This all begs the question of what the model uses as input for the Amazon discharge. Nothing is said about this in section 2.4. Similar comment applies to lines 495-498.
418-419. "more saline" I do not see this. The anomalies shown in Figure 3 are about the same for all the datasets except HYCOM.
475. "land contamination" RSS70 has the same issue with land contamination. The reduced smoothing is much more likely.
488-489. "respond...freshwater" Alternate wording suggestion: "resolve fresh mesoscale features"
495. I'm not sure it's the climatology that is the issue, as elaborated in the following sentences.
543. Wait, I thought there were three R/Vs (lines 132-134).
Section 5.2. This material is stuck onto at the end of the conclusions section without a proper introduction of where the data come from, or motivation as to why the analysis is being carried out. It is completely out of place and should be deleted, or moved into the main part of the paper.
585-594. To reduce wording and confusion, the authors can just give one DOI link for each dataset they use, e.g. https://doi.org/10.5067/SDRON-ATOM0
588-590. These DOIs lead to L2 datasets. The authors also need to link to the L3 datasets they used, and clearly state which is which.
588. This link is to an Oceanography article, not the dataset itself.
626-627. This reference is incomplete. I do not know what this is. Ditto line 652, 670-671.
Lines 684-689. What are these references? Is it software? Documentation? At the least, the authors should provide DOIs to make them easier to find. I suppose I could google the author list, but why not make it easy for the reader. I found this, https://zenodo.org/record/4774304, for example for reference 38. It has an "interp" method in it (line 277), but no discussion of what this method does or how it works.
Author Response
Our response is in the attached PDF.

Round 3
Reviewer 2 Report
This is my third review of this paper. The authors have responded to many of my previous comments positively. However, they did not do much to tighten up the manuscript, and there are still a lot of issues with they way they describe their results. There are many long passages that are unnecessary (e.g. lines 199-201, 202-211, 227-243, etc.). The authors need to go through their paper word-by-word, sentence-by-sentence, and make sure that every word is both necessary and sufficient to tell the story they want to tell. They also need to critically examine every word they write to verify that it accurately describes the figures and tables. They do have results that are valuable to the community and should be communicated, but there is still much work to be done to make it publishable. See individual comments below.
17-18. They don't just "include" these products. They are the only ones being used in the paper. Also please rewrite this sentence to make it clear just exactly what products are being tested, and that the tests are done at both L2 and L3.
66. "too coarse to..." what?
The BdT notation is still present in Figure 2.
Figure 2 is still pixellated and poor quality.
173. "data are returned"
Figure 3. The caption (or a legend) should indicate which saildrone is which color. I guess they double back on themselves sometimes The tracks seem to have three endpoints. The authors might add some kind of symbol showing the sites where each track begins and ends.
187. 48W, 60W.
193. Maybe the tropical Atlantic, or southern North Atlantic
199-201. This information is in the data availability statement and is redundant. Delete.
202-211. Most of this is superfluous, all but the final sentence.
227-243. Much of this is unnecessary and can be deleted.
244. Which two? The RSS L2 and L3, or the JPL L2 and L3?
250. Yes, but the data the authors are using here are not from anywhere near 40N.
251-253. Is this a problem is this region? If so, how did the authors deal with it? If not, why mention it?
270. Fournier et al
302. "simulate" I have not looked at these two references... But I would guess that the fluxes are imposed on the model, not simulated within the model. The authors can correct me.
324. "this" Which dataset does this refer to? This sentence needs some clarification, or maybe should be broken into two.
330. What does this mean? Why did they install the RBR instruments if they are not recommended? And who is the personal communication from?
334. "a constant calibration offset" So the CTDs must have been factory-calibrated differently for each instrument. (line 207) This seems strange as in theory they should both have been calibrated against standard seawater.
344. "17-Jan" I guess the comparison started two days after the saildrones were deployed (line 186). Ditto line 366.
362. I cannot figure out which JPL product is being displayed, the L2 or L3. Both products are included in Table 2. Ditto figure 5.
369-370. "the 3-hour HYCOM product was collocated and interpolated" This makes sense for the L2 values, i.e. the model was interpolated to the time and location of the satellite pass. I'm guessing that for the 8-day HYCOM salinities some kind of average was formulated. The details need to be provided - or maybe I missed it.
374-376. This simply repeats the table caption. Delete.
388. "0.03" I cannot see in Table 2 where any of the STD numbers is less than this. On the next line, the SMAP STD numbers never approach 0.1. The "in difference" wording is confusing. The authors need to consider what they are trying to say.
390-391. The correlation coefficients are between the saildrones and satellite products, so this statement makes no sense.
392-394. This is true, but it is hard to tell how significant it is. We are not given any sort of confidence limits for these correlation coefficients.
406. I am not sure I would call what HYCOM puts out "observations". "Estimates" maybe. Can the authors explain the different character of the comparison in Figure 5d vs. the rest of the panels in Figure 5. It looks like HYCOM would have a much smaller RMSD than the other products here, but that does not match what is in Table 2.
416-418. I don't see what this refers to. Both the saildrones and SMAP show variability. It's unclear that SMAP shows less of it.
422. "slightly more saline" Again, I cannot see this. The JPL and RSS70 products seem to be very similar. The RSS40 is very noisy.
424. The RSS40 product is more variable in mid-ocean. Do the authors have some indication that this variability is not real? If it is real, then the RSS40 product really does have better spatial resolution.
Figure 6. I do not understand the purpose of showing the entire set of saildrone records in this figure (15-January - 2-March), when the rest of the figure is only valid for 17-Feb +-4 days. Also, the color scale the authors have chosen for this figure (and Fig. 7) is not ideal. It washes out almost all mesoscale variability. The HYCOM panel especially looks almost completely uniform.
Figure 7. This figure is superfluous as the point the authors are trying to make with it (that HYCOM depicts the Amazon plume poorly) can be easily seen in Figure 6.
452. "unique" I'm not sure what the authors mean by this word. Suggest deleting it.
Figure 8. There is a color bar in panels c) and d) that is not described in the caption. I can guess what it might be, but should not have to guess. The caption should use the word "stickplot" in describing these panels somewhere.
507. Figs. 3 and 7 say nothing about a temporal offset.
531. "large spatial resolution" This contradicts line 295. The version of HYCOM the authors are using has a 0.08deg resolution (line 304) - not sure if this is "large" or "fine", but it seems plenty adequate to resolve the fresh tongue which has a scale of ~200 km (Fig. 8). As explained subsequently, the inability to depict the fresh tongue is not the result of limited model resolution.
Section 5.1 is quite repetitive from earlier. Is it necessary?
581-597. This information on the sampling and calibration of TSG data on EUREC4A-OA/ATOMIC's R/Vs belongs in section 2. Or else, since these data are never used in this study, should be deleted. Instead of focusing on how the TSG data are sampled and calibrated, the authors should focus on how they might be used in this region to further validate the satellite data, i.e. the last couple of sentences.
600. Is the version of HYCOM the authors used the "default version". What does this mean?
598-602. This seems like a good idea. Are there other such experiments in the literature the authors can point to as examples?